# Fine Segmentation of Chinese Character Strokes Based on Coordinate Awareness and Enhanced BiFPN

**DOI:** 10.3390/s24113480

**Published:** 2024-05-28

**Authors:** Henghui Mo, Linjing Wei

**Affiliations:** College of Information Science and Technology, Gansu Agricultural University, Lanzhou 730070, China; mohh@st.gsau.edu.cn

**Keywords:** Chinese character stroke segmentation, instance segmentation, YOLOv8n-seg, attention mechanism, BiFPN, Shape-IoU, Grad-CAM++

## Abstract

Considering the complex structure of Chinese characters, particularly the connections and intersections between strokes, there are challenges in low accuracy of Chinese character stroke extraction and recognition, as well as unclear segmentation. This study builds upon the YOLOv8n-seg model to propose the YOLOv8n-seg-CAA-BiFPN Chinese character stroke fine segmentation model. The proposed Coordinate-Aware Attention mechanism (CAA) divides the backbone network input feature map into four parts, applying different weights for horizontal, vertical, and channel attention to compute and fuse key information, thus capturing the contextual regularity of closely arranged stroke positions. The network’s neck integrates an enhanced weighted bi-directional feature pyramid network (BiFPN), enhancing the fusion effect for features of strokes of various sizes. The Shape-IoU loss function is adopted in place of the traditional CIoU loss function, focusing on the shape and scale of stroke bounding boxes to optimize the bounding box regression process. Finally, the Grad-CAM++ technique is used to generate heatmaps of segmentation predictions, facilitating the visualization of effective features and a deeper understanding of the model’s focus areas. Trained and tested on the public Chinese character stroke datasets CCSE-Kai and CCSE-HW, the model achieves an average accuracy of 84.71%, an average recall rate of 83.65%, and a mean average precision of 80.11%. Compared to the original YOLOv8n-seg and existing mainstream segmentation models like SegFormer, BiSeNetV2, and Mask R-CNN, the average accuracy improved by 3.50%, 4.35%, 10.56%, and 22.05%, respectively; the average recall rates improved by 4.42%, 9.32%, 15.64%, and 24.92%, respectively; and the mean average precision improved by 3.11%, 4.15%, 8.02%, and 19.33%, respectively. The results demonstrate that the YOLOv8n-seg-CAA-BiFPN network can accurately achieve Chinese character stroke segmentation.

## 1. Introduction

In the field of digital processing of Chinese characters, stroke segmentation technology is fundamental for achieving high-quality recognition of Chinese characters [1], analysis of calligraphy art [2], and font design [3]. However, the complex structure of Chinese characters, particularly the interconnections and intersections among strokes, has long posed significant challenges to stroke segmentation. The complexity and diversity in the structure of Chinese characters, especially the interconnections and intersections among strokes, have greatly challenged the extraction of strokes. Traditional methods of stroke extraction can generally be divided into two categories: unsupervised methods based on image processing techniques and methods based on data-driven or model approaches.

In early research, many scholars attempted to address the problem of Chinese character stroke extraction using various image processing techniques. For instance, by analyzing the contours of Chinese characters, the areas within the boundaries were transformed into triangular meshes using constrained Delaunay triangulation, and target strokes were extracted through the synthesis of sub-strokes based on continuity analysis [4]. Additionally, studies have integrated bottom-up and top-down approaches and applied Markov Random Field models for stroke extraction, or combined Canny edge detection and constrained Delaunay triangulation grids, as well as PBOD techniques, to achieve Chinese character stroke segmentation [5,6]. These methods have achieved some success in extracting basic strokes, but they often incorrectly segment composite strokes into several basic strokes.

To overcome these issues, subsequent research began to incorporate annotated datasets, using the GB2312 standard script library as reference data [7,8]. Initially, the character forms were skeletonized, and then point set registration algorithms, such as the Coherent Point Drift method, were used to match the skeleton points of the target characters with those of the reference characters [9]. There were also approaches using skeleton crawlers to extract strokes [10]. Although these studies have made progress in the accuracy of stroke segmentation, they still have limitations, particularly when dealing with complex scenarios in calligraphic fonts or diverse handwritten styles, where these methods often fail due to a lack of understanding of advanced semantic information.

Against this backdrop, the rapid development of deep learning technologies has provided new approaches to address these issues. Particularly in the fields of image processing and computer vision, deep learning has demonstrated strong representational and approximation capabilities [11], making it possible to perform fine segmentation of Chinese character strokes. Wang et al. [12] proposed a stroke extraction framework using an improved Deepstroke semantic segmentation model and tabu search for stroke segmentation and order recognition. Bi Fukun et al. [13] introduced a Generative Adversarial Network (SSGAN) for calligraphy image stroke segmentation, which significantly improved segmentation accuracy through multi-stroke tensor training strategies and a ResU-Net structure. Zhang et al. [14] began using Conditional Generative Adversarial Nets (CGAN) to generate calligraphy strokes, although limited to single-character generation. Li Meng et al. [15] effectively improved stroke separation and matching accuracy using technologies like the Structural Deformable Image Registration Network (SDNet). However, despite significant progress in image segmentation tasks, deep learning models still face challenges in effectively capturing complex stroke structures and details when dealing with Chinese character stroke segmentation.

Despite some progress made by traditional methods and preliminary deep learning models, multiple challenges still persist in practical applications. Particularly, issues such as the clarity of stroke segmentation, the accuracy of edge contours, and adaptability to different font styles, as well as overfitting and weak generalization abilities due to imbalanced sample categories, often make it difficult for existing methods to meet the requirements of high precision and efficiency. Therefore, based on the high segmentation accuracy, multi-scale target segmentation effectiveness, and fast segmentation speed of the YOLOv8n-seg (You Only Look Once v8 Nano Segment) model [16], this study proposes a YOLOv8n-seg-CAA-BiFPN network to achieve fine segmentation of Chinese character strokes, thereby paving new pathways for the application of deep learning in Chinese character design and computer-aided design technologies.

## 2. Materials and Methods

### 2.1. YOLOv8n-seg Network Structure Improvement

The YOLOv8n-seg primarily consists of Input, Backbone, Neck, and Head components. The Input module adjusts the size of the input image to meet the network’s requirements. The Backbone module incorporates improved CSP ideas, replacing the C3 module in YOLOv5 with a C2f module, and retains the SPPF module to enhance residual learning capabilities. The Neck part draws from the PAN structure, optimizing the construction of the feature pyramid and removing some unnecessary convolutional structures from previous versions to make feature fusion more efficient. The Head part decouples the classification and regression tasks and abandons the anchor-based design in favor of an anchor-free approach, enhancing the model’s flexibility and accuracy. However, in the field of Chinese character font segmentation, research and application face a series of unique challenges. These challenges include, but are not limited to, the recognition difficulties caused by differences between different font styles, the complexity of Chinese characters’ structure, and the subtle differences in strokes affecting precise segmentation, as well as font deformations in practical applications disturbing the segmentation effects. Moreover, the diversity of handwritten and artistic fonts and the common occurrence of strokes crossing and overlapping in these fonts significantly increase the difficulty of segmentation tasks. These issues were not adequately addressed in the original YOLOv8n-seg architecture, which could lead to decreased detection performance.

The network framework proposed in this study for the fine segmentation of Chinese character strokes is illustrated in Figure 1. The process begins by collecting images of Chinese characters in both Kai style and handwritten forms as image data, resizing them to a resolution of 128 × 128. Image augmentation techniques are employed to expand and enhance the diversity and complexity of the dataset. In the detection network, pre-processed images and their corresponding labels are trained to obtain model weight files. During the training process, specific loss functions and optimizers are used to ensure optimal performance of the model for the designated tasks. Lastly, the trained model weight files are used to validate and test on test images to evaluate the model’s performance and accuracy. Training metrics such as accuracy, recall rate, and mean average precision are recorded to comprehensively assess the performance of the model. The entire process is designed to efficiently and accurately segment strokes of Chinese characters in various styles, ensuring stability in subsequent Chinese character design and computer-aided design technologies.

### 2.2. Coordinate-Aware Attention (CAA)

In tackling the challenges of fine segmentation of Chinese character strokes, the YOLOv8n-seg model encounters difficulties in capturing tightly arranged strokes and their contextual patterns, especially challenges such as diverse stroke shapes, low prominence, and mutual occlusion. To address these issues, this study introduces the Coordinate Aware Attention Mechanism (CAA), whose network structure is shown in Figure 2, and is implemented behind the bottleneck modules in the second, third, and fourth C2f modules of the Backbone network. This enables the model to adaptively adjust its focus by precisely calculating and integrating attention weights across different spatial dimensions (vertical and horizontal) and channel dimensions, thereby enhancing the model’s ability to recognize the shape, size, and positional context of Chinese character strokes. This network architecture leverages the modeling potential of differentiated attention weights and optimizes the dimensionality of information querying modules, ensuring effective management of computational resources while improving the model’s capability for fine segmentation of Chinese character strokes in complex environments.

The Coordinate Aware Attention mechanism primarily performs information fusion across spatial dimensions, channel dimensions, and historical orientations. In the feature map, different positions carry varying degrees of importance; thus, the Coordinate Aware Attention mechanism refines and differentiates the input feature map to precisely capture and merge information both horizontally and vertically, as well as across channels. Initially, CAA divides the input feature map along the channel dimension into four independent parts. The first and second parts are for spatial dimension operations, performing adaptive average pooling in vertical and horizontal directions, respectively. This step is followed by a feature information concatenation operation aimed at capturing sensitivity to different directions. These operations are then shuffled using a 1 × 1 convolution module to produce encoded feature representations for vertical (*F*_*x**c*_) and horizontal (*F*_*y**c*_) directions. Another part *F*_*x**y*_ continues to enhance features through a 3 × 3 convolution module, considering different scale receptive fields. The Sigmoid function generates attention weights (*F*_*s**x**y*_), and the encoded features *F*_*x**c*_ and *F*_*y**c*_ are then weighted by *F*_*x**w*_ and *F*_*y**w*_, respectively. These weights are activated by another Sigmoid function to produce differentiated weight values for the X and Y coordinates. These weights are then multiplied element-wise with the input features, resulting in the final horizontal (*F*_*a**x*_) and vertical (*F*_*a**y*_) attention maps. The process is outlined as follows:(1)$$F_{1},F_{2},F_{3},F_{4}=Split(f)$$
(2)$$F_{x},F_{y}=Xavg(F_{1}),P(Yavg(F_{2}))$$
(3)$$F_{xy}=CBS_{1*1}(Concat(F_{x},F_{y}))$$
(4)$$F_{xc},F_{yc}=Split(F_{xy})$$
(5)$$F_{exy}=CBS_{3*3}(F_{xy})$$
(6)$$F_{sxy}=\partial (F_{exy})$$
(7)$$F_{xw},F_{yw}=Split(F_{sxy})$$
(8)$$F_{ax},F_{ay}=F_{1}⊗(\partial (F_{xc}⊙F_{xw})),F_{2}⊗(P(\partial (F_{yc}⊙F_{yw})))$$

In this context, *f* represents the input feature map; “Split” refers to segmenting the feature map along the channel dimension; *X*_avg_ and *Y*_avg_, respectively, represent performing average pooling operations along the X and Y directions; “Concat” signifies the concatenation operation along the channel direction; *C*BS_*n***n*_ indicates the use of an *n* × *n* convolution module; ∂ represents the Sigmoid function; ⊙ denotes multiplication; and ⊗ represents the dot product operation.

The third part is the channel dimension branch, which captures global channel information through global average pooling, G_av_. It uses the attention weights from the spatial dimensions, *F*_*s**x**y*_, to perform an averaging operation, integrating weight information. This integrated data is then dot-multiplied with the global channel information, blending spatial and channel information. Differentiated attention weights are generated using the Sigmoid function. These weights not only adjust for vertical and horizontal feature maps but also finely weight the features in the channel dimension, achieving efficient fusion and reshaping of features. This is then element-wise multiplied with the input features to obtain the final channel-direction attention map, *F*_*a**c*ℎ_. The process is depicted as follows in Formula (9):(9)$$F_{ach}=\partial (Gav(F_{3})⊙F_{sxy})$$

The fourth part is the historical orientation branch, which utilizes a 1 × 1 convolution to perform residual connections, inheriting historical orientation information. This not only preserves long-term dependencies but also provides the model with an effective method to integrate historical and current layer information, thereby achieving a deeper understanding of complex Chinese character structures. The resulting historical orientation attention map is denoted as *F*_*a*old_. The process is represented in Formula (10):(10)$$F_{aold}=CBS_{1*1}(F_{4})$$

Ultimately, the CAA refines and outputs a feature map containing rich coordinate information and differentiated attention weights, *F*_CAA_, through a 1 × 1 convolution and Sigmoid activation function. This is illustrated in Formula (11):(11)$$F_{CAA}=CBS_{1*1}(Concat(F_{ax},F_{ay},F_{ach},F_{aold}))$$

The CAA not only delves deeply into the spatial structure of the input features but also ingeniously integrates information across spatial, channel, and historical dimensions. This significantly enhances the model’s perception and handling of detailed features. Particularly in tasks involving fine segmentation of Chinese character strokes, which heavily rely on spatial information, the CAA demonstrates its superiority through the efficient integration of three-dimensional information. This approach also provides new insights and technical pathways for future research in deep neural networks, particularly in capturing and processing fine-grained spatial information.

### 2.3. Enhanced BiFPN

In the traditional YOLOv8n-seg, the Neck part borrows from the PAN structure [17], which strengthens the connection between lower and higher-level features, thus enhancing the richness and representational ability of the feature map. However, when dealing with extremely fine features or highly complex scene segmentation tasks, relying solely on the PAN structure may lead to insufficient feature fusion in the YOLOv8n-seg model [18]. To address this, some studies have proposed the use of a BiFPN connection structure to enhance the feature fusion capabilities of YOLOv8n-seg. The BiFPN structure builds upon PAN, introducing new channels between the same feature’s input and output nodes while removing single input edges that contribute little to feature fusion or extraction.

However, in the standard BiFPN design of YOLOv8n-seg, feature fusion is primarily limited to higher-level feature maps from P5 to P3, which is generally sufficient when dealing with objects across a broad scale. However, for finely detailed stroke features, such as the edges of written characters or small objects, traditional BiFPN may not provide adequate feature resolution. These higher-level feature maps might lack necessary spatial details, thus limiting the model’s performance in tasks requiring high precision.

To address these issues, this study proposes an enhanced BiFPN structure as a replacement for the neck architecture of YOLOv8n-seg, as shown in Figure 3. By extending feature fusion down to the P2 layer, the model can handle fine strokes and complex touches in Chinese characters, restoring details that are easily lost in higher-level feature maps. Before performing skip connections, a CBS layer (Convolution-BatchNorm-Swish) is used to enhance the features at each fusion node, improving the representation capacity and sensitivity to detail changes. Particularly, an enhanced weighted fusion strategy is employed, using the ReLU activation function to ensure the non-negativity of weights, followed by normalization to maintain the total weight sum to one, ensuring that contributions from each feature layer are proportionally fused. The enhanced feature fusion strategy dynamically allocates the weights of feature maps from various network levels, thus optimizing the information flow. Specifically, the fusion process of BiFPN can be described in detail through the following formulas:

First, each feature map x_i_ is assigned an initial weight w_i_, and the ReLU function is applied to ensure the weights are non-negative:(12)wi′=ReLU(wi)

Next, to ensure that the sum of weights equals one, the weights activated by ReLU, w^′^_i_, are normalized:(13)$$w_{i}^{\prime \prime }=\frac{w_{i}^{\prime }}{\sum_{j=1}^{n}w_{j}^{\prime }+\varepsilon }$$

Finally, the normalized weights w^″^_i_ are used to compute a weighted sum of the feature maps x_i_ to obtain the final fused feature map *Y*:(14)$$Y=\sum_{i=1}^{n}w_{i}^{\prime \prime }\cdot x_{i}$$

The enhanced BiFPN structure ensures the preservation of detail information in higher-level feature maps, while also promoting effective integration between lower-level details and higher-level semantics, providing strong technical support for the precise segmentation of complex Chinese character strokes.

### 2.4. Shape-IoU Loss Function

The purpose of bounding box regression is to fine-tune the detection model’s output candidate boxes to maximize their overlap with the true boundaries of the target objects. Therefore, the Intersection over Union (IoU) serves as a metric to measure the degree of overlap between the predicted and true boxes, as shown in Formula (15):(15)$$IoU=\frac{\left|B^{pred}\cap B^{gt}\right|}{\left|B^{pred}\cup B^{gt}\right|}$$

The YOLOv8n-seg model employs the Complete Intersection over Union (CIoU) metric as part of its bounding box regression [19], as illustrated in Formulas (16)–(18):(16)$$v=\frac{4}{\pi }(\arctan\frac{w^{gt}}{h^{gt}}-\arctan\frac{w^{pred}}{h^{pred}})$$
(17)$$\alpha =\frac{v}{(1-IoU)+v}$$
(18)$$CIoU=IoU-(\frac{\rho^{2}(B^{pred},B^{gt})}{c^{2}}+\alpha v)$$

In these formulas, *v* represents a correction factor, *α* is a balance parameter, and *β* represents the Euclidean distance between the centers of the predicted and true boxes.

CIoU in bounding box regression integrates overlap area, center distance, and aspect ratio to optimize the prediction box. However, its strong sensitivity to aspect ratios can result in lower scores for specific target shapes, potentially affecting detection performance. Additionally, a large deviation in the center point can cause the model to overly focus on position accuracy while neglecting adjustments in the overlapping areas. This may require further balance and refinement in distance estimation.

Therefore, this study adopts the Shape-IoU loss function [20] as the bounding box regression method for the YOLOv8n-seg-CAA-BiFPN model. Shape-IoU optimizes the shape and scale of bounding boxes by integrating shape adaptability loss and scale adaptability penalties, enhancing the model’s adaptability and accuracy on targets of varying sizes, particularly for fine and variably shaped targets like Chinese character strokes. This improvement not only addresses the weaknesses in CIoU’s sensitivity but also its inadequacy in adapting well to irregular shapes or small-scale strokes. Specifically, the principles of the Shape-IoU loss are depicted in Formulas (19)–(23):(19)$$w_{w}=\frac{2\times {\left(w^{gt}\right)}^{s}}{{\left(w^{gt}\right)}^{s}+{\left(h^{gt}\right)}^{s}}$$
(20)$$h_{h}=\frac{2\times {\left(h^{gt}\right)}^{s}}{{\left(w^{gt}\right)}^{s}+{\left(h^{gt}\right)}^{s}}$$
(21)$$d^{s}=h_{h}\times \frac{{\left(x_{c}-x_{c}^{gt}\right)}^{2}}{c^{2}}+w_{w}\times \frac{{\left(y_{c}-y_{c}^{gt}\right)}^{2}}{c^{2}}$$
(22)$$\Omega^{s}={\sum_{t=w,h}\left(1-e^{w_{t}}\right)}^{\theta },\theta =4$$
(23)$$\left\{\begin{array}{l} \omega_{w}=h_{h}\times \frac{\left|w-w^{gt}\right|}{\max(w,w^{gt})}\\ \omega_{h}=w_{w}\times \frac{\left|h-h^{gt}\right|}{\max(h,h^{gt})} \end{array}\right.$$

In this context, *s* represents a scale factor, which is related to the scale of targets within the dataset. *W*_*w*_ and ℎ_ℎ_ are the weight coefficients for the horizontal and vertical directions, respectively, and their values are related to the shape of the ground truth (GT) box. The corresponding bounding box regression loss is represented in Formula (24):(24)$$L_{Shape-IoU}=1-f_{IoU}+d^{s}+0.5\times \Omega^{s}$$

In this formula: *f*_*I**o**U*_ denotes the basic Intersection over Union, *d**s* represents the shape adaptability loss, Ω^s^ signifies the scale adaptability penalty.

As shown in Figure 4, Shape-IoU focuses on adjusting the predicted bounding box to better match the size and shape of the GT box. By quantifying shape adaptability and scale adaptability, it provides a more comprehensive method for bounding box regression.

## 3. Model Training and Result Analysis

### 3.1. Experimental Environment Configuration

To more vividly showcase the achievements of Chinese character stroke segmentation technology, this study has developed an interactive educational platform. This platform operates in a high-performance computing environment, equipped with a GeForce RTX 3090 Ti graphics card and an Intel i9-13900HQ@5.40GHz CPU. The graphics card is manufactured by NVIDIA, based in Santa Clara, California, USA. The CPU is produced by Intel Corporation, located in Santa Clara, California, USA. The platform runs on the Ubuntu 18.04 LTS operating system, developed by Canonical Ltd., headquartered in London, UK, uses Pytorch 1.7.1 as the deep learning framework, and utilizes Pycharm 2020.3 for development. Image processing is conducted using OpenCV 3.4.6, and the programming was conducted in Python 3.7.0.

The improved YOLOv8n-seg network model is specifically designed for challenging tasks of fine-grained segmentation of Chinese character strokes. Users can not only observe precise segmentation results but can also engage in real-time learning and feedback through interactive operations. The user interface of the Chinese character stroke segmentation system is shown in Figure 5.

### 3.2. Dataset Construction

The experimental data are derived from the publicly available Chinese character stroke datasets CCSE-Kai and CCSE-HW, proposed by the team from South China University of Technology [21]. These datasets include styles of Kai script and handwritten fonts, respectively. The CCSE-Kai dataset contains 7667 training instances, 919 validation instances, and 937 testing instances. The CCSE-HW dataset includes 6131 training instances, 768 validation instances, and 729 testing instances, ensuring robust data support and effective validation of experimental results. Figure 6 shows the number of annotated instances per category.

For image segmentation tasks, the CCSE datasets represent fine-grained challenges. As shown in Figure 7, Chinese character strokes inherently possess high complexity and detail, such as dots, downward strokes, and rightward strokes. Some fine strokes or intertwined parts may be lost or misclassified during the segmentation process. Additionally, significant differences in stroke thickness and curvature among different font styles increase the difficulty for a single model to handle multiple styles effectively.

### 3.3. Model Training Parameter Settings

In this study, the Stochastic Gradient Descent (SGD) optimizer [22] is used to optimize training loss, with a momentum setting of 0.937 to enhance the stability and efficiency of the optimization. Regarding the training strategy of the model, an initial learning rate of 0.01 is set for the first 200 epochs, aimed at quickly approaching a reasonable solution for the model. As the model progressively approaches its optimal state, the learning rate is precisely reduced to 0.0001 for the subsequent 100 epochs to finely adjust performance. Additionally, to prevent the phenomenon of overfitting, a weight decay of 0.0005 (L2 regularization) is introduced. The model employs a warm-up phase where a lower learning rate is used in the initial three epochs and gradually increased to the predetermined value to address difficulties in training due to unstable model weight initialization.

### 3.4. Experimental Results and Analysis

To verify the effectiveness of the algorithm proposed in this study, we selected the Mask R-CNN [23] from the CCSE paper and other mainstream segmentation models like BiSeNetV2 [24], SegFormer [25], YOLOv8n-seg [16], and YOLOv8n-seg-CAA-BiFPN models.

Figure 8, Figure 9 and Figure 10 show line charts depicting the trends in average accuracy, recall rates, and mean average precision for the five models during the training process. From a detailed numerical analysis, Mask R-CNN showed considerable fluctuation during training, struggling to quickly learn semantic information between strokes for segmentation decisions, similar to the relatively lower accuracy of the traditional encoder structure model BiSeNetV2. SegFormer, based on the Transformer model structure, was able to better grasp global information and understand the arrangement and combination of strokes. After about 200 rounds of iteration, the average accuracies for SegFormer and YOLOv8n-seg stabilized at around 80% and 81%, respectively. The YOLOv8n-seg-CAA-BiFPN model, with the introduction of the Coordinate Aware Attention Mechanism in its Backbone, improved the overall segmentation model’s understanding of different font styles and their positional information sensing abilities, gradually increasing the average accuracy to about 84%, and finally stabilizing at approximately 84.7%.

In terms of average recall rates, due to the limitations of Mask R-CNN’s Region Proposal Network in fine-grained tasks and the simplified design of BiSeNetV2, which balances speed and accuracy, both had lower recall rates compared to other models. SegFormer and YOLOv8n-seg had an average recall rate of about 77% in segmenting positive samples. The YOLOv8n-seg-CAA-BiFPN model, with the Neck incorporating an enhanced BiFPN through refined feature fusion strategies and understanding of lower layer features, fully grasped the hierarchical features of different sized strokes and optimized the capture of small overlapping stroke information. Thus, compared to the original model, the improved model increased the recall rate by about 4%, reaching 83.65%.

The YOLOv8n-seg-CAA-BiFPN model also performed excellently in mean average precision comparisons. By optimizing the bounding box regression with the Shape-IoU loss function focusing on the shape and scale of stroke borders, the model significantly enhanced its prediction accuracy. In handling fine segmentation tasks of Chinese character strokes, it effectively improved the model’s mean average precision.

A comprehensive analysis shows that the improved YOLOv8n-seg model significantly enhanced both performance and stability in fine segmentation tasks of Chinese character strokes.

Data analysis from Table 1 shows that the YOLOv8n-seg-CAA-BiFPN model outperforms Mask R-CNN, BiSeNetV2, SegFormer, and the original YOLOv8n-seg in the fine segmentation task of Chinese character strokes. It also shows advantages in the smaller size of the weight files, highlighting the superior performance of the YOLOv8n-seg-CAA-BiFPN in handling complex tasks of fine segmentation of Chinese character strokes.

### 3.5. Results of Ablation Experiment

To better validate the effectiveness of the improved algorithm over the original, three different network configurations were designed for ablation experiments, all using the same CCSE Chinese character stroke dataset, batch sizes, and training cycles.

According to the data in Table 2, the most significant improvement was observed when the CAA module was individually added to the YOLOv8n-seg model, increasing the average accuracy by approximately 1.91%, the average recall rate by about 1.78%, and the mean average precision by about 1.9%. After implementing all the improvement strategies on the YOLOv8n-seg model, the average accuracy reached 84.71%, the recall rate was 84.71%, and the mean average precision was 83.65%, with a mean average precision value of 80.11%. Furthermore, while maintaining high recognition accuracy, the model achieved rapid recognition in 0.027 s, demonstrating that this integrated strategy effectively enhances the overall performance of fine-grained segmentation of Chinese character strokes while ensuring efficiency. To verify the advantages of this study’s algorithm, the improved YOLOv8n-seg-CAA-BiFPN model was deployed to the detection platform. The model was then used to recognize data from the CCSE test set, and five representative sets of detection results were selected for comparative validation.

As shown in Figure 11, the segmentation results based on traditional models like Mask R-CNN and BiSeNet V2, while capable of identifying most strokes in complex Chinese character stroke segmentation tasks, exhibit some degree of missed and incorrect detections. This may be due to limitations in these models’ ability to capture the contrast between detailed strokes and the background. In comparison, SegFormer shows certain improvements in segmenting detailed strokes, likely due to its better spatial resolution capabilities; however, it still struggles with boundary blur and overlapping strokes in cases of adherent strokes.

The YOLOv8n-seg model demonstrates a certain level of recognition effectiveness, but there is room for improvement in resolution and accurate boundary delineation among closely arranged strokes. This indicates that although YOLOv8n-seg performs well in general stroke segmentation tasks, it might not yet be fully optimized for more refined segmentation, such as understanding the contextual rules of stroke positioning.

In the results of the YOLOv8n-seg-CAA-BiFPN model, clearer stroke boundaries and prominent shape recognition are displayed. This emphasizes the effectiveness of the Coordinate Aware Attention Mechanism in capturing the contextual rules of stroke positioning, and the importance of the enhanced BiFPN in improving the feature fusion effects of strokes of various sizes. Additionally, the Shape-IoU loss function, which focuses more on the shape and scale of the strokes during bounding box regression, helps enhance the model’s performance in fine-grained stroke segmentation tasks.

### 3.6. Performance Analysis of Attention Mechanism

To comprehensively assess the performance of the Coordinate Aware Attention (CAA) in the YOLOv8n-seg model, this study conducted an in-depth comparison of the CAA with several popular attention mechanism improvement methods used in other similar research papers and carries out ablation experiments for analysis. The purpose of these experiments was to analyze the performance differences of these various attention mechanisms in segmentation tasks involving diverse stroke shapes, low prominence, and overlapping features. Specifically, the experiments involved integrating the respective attention mechanisms immediately following the Bottleneck modules in the second, third, and fourth C2f modules of the Backbone, creating configurations like YOLOv8n+SE, YOLOv8n+CA, YOLOv8n+CBAM, and comparing these with YOLOv8n+CAA1, YOLOv8n+CAA2, and YOLOv8n+CAA3, where CAA is integrated solely in the second, third, and fourth C2f modules, respectively, as well as YOLOv8n+CAA where CAA is added across all specified modules. Table 3 displays the performance metrics for these eight configurations.

The experimental results show that integrating the CAA mechanism individually at different parts of the YOLOv8n-seg model improved the average precision in each case, with the most significant performance enhancement observed when CAA was implemented throughout the model. Specifically, YOLOv8n+CAA achieved an average precision of 78.90%, an improvement of 1.9% over the original YOLOv8n-seg model. Although this integration resulted in a slight increase in model size and a longer processing time, these changes are considered acceptable adjustments made to significantly enhance segmentation performance. This indicates the effectiveness of the CAA mechanism in handling Chinese character segmentation tasks, particularly in processing features of low prominence and fine detail.

### 3.7. Enhanced BiFPN Analysis

To assess the effectiveness of the Enhanced BiFPN module within the YOLOv8n-seg network, this study conducted performance evaluations through ablation experiments involving different BiFPN configurations in the Neck of the YOLOv8n-seg network.

As shown in Table 4, YOLOv8n+BiFPN denotes the application of the original BiFPN structure in the Neck; YOLOv8n+Enhanced BiFPN denotes the integration of the proposed enhanced BiFPN with the Neck. The results indicate that compared to the original BiFPN, the Enhanced BiFPN, while maintaining the model size and processing time efficiency, can improve the network’s mean average precision to 78.54%. Additionally, compared to the original YOLOv8n-seg Neck structure, it not only improves average precision but also achieves more notable reductions in model size by 2.0 MB and enhances detection speed by 0.004 s. This balance between performance and real-time capability makes the enhancements more suitable for practical application needs.

### 3.8. Heat Map Analysis

Heat map analysis is an important tool for enhancing the interpretability of object detection models, especially when dealing with highly complex visual tasks such as the fine segmentation of Chinese character strokes. Grad-CAM++ is an advanced visualization technique that improves upon the standard Grad-CAM method [26]. It allows for more precise localization of the image areas that the model relies on when making decisions by calculating the gradients of the output from the last convolutional layer and using these gradients to weight the channels in the feature maps to generate heat maps. Compared to Grad-CAM, Grad-CAM++ considers higher-order derivative information when calculating weights, which enables it to more finely capture and emphasize areas that have a significant impact on decisions.

In the context of fine segmentation of Chinese character strokes, this means that Grad-CAM++ can reveal how the model differentiates subtle variations in strokes and the complex intersections and overlapping areas between strokes. For example, in this study, Grad-CAM++ technology was used to analyze the detection heat maps of the YOLOv8n-seg and YOLOv8n-seg-CAA-BiFPN models. The results are shown in Figure 12.

The heat map comparison in Figure 12 clearly reveals the differences in the areas of focus between the two models on the task of fine-grained segmentation of Chinese character strokes. In the YOLOv8n-seg model, the heat distribution is relatively scattered, indicating a deficiency in recognizing key stroke details, especially in parts of Chinese characters with complex structures. On the other hand, the YOLOv8n-seg-CAA-BiFPN model displays more concentrated hotspots, particularly at the junctions and intersections of strokes, which are typically crucial areas for determining the structure of Chinese characters. This model reduces attention to non-critical areas, especially in complex backgrounds, demonstrating its strong capabilities in capturing details and eliminating background noise. Overall, the performance of the YOLOv8n-seg-CAA-BiFPN model in terms of recognition accuracy and focus is superior to that of the standard YOLOv8n-seg, proving its suitability and efficiency in the task of fine-grained segmentation of Chinese character strokes.

## 4. Conclusions

In response to the high complexity and subtle variations in Chinese character stroke segmentation, this study proposed an improved scheme based on the YOLOv8n-seg model, named YOLOv8n-seg-CAA-BiFPN, to enhance the accuracy and adaptability of traditional models in segmenting Chinese character strokes. This improvement is particularly evident in defining edges and processing diverse fonts, as well as in mitigating overfitting issues caused by sample imbalance. The experiments led to the following conclusions:

The proposed Coordinate Aware Attention (CAA) mechanism optimized the capture of Chinese character stroke features and enhanced the model’s understanding of the positional context of strokes, especially in scenarios with complex structures and tightly arranged strokes. The Enhanced BiFPN effectively fused features of strokes of various sizes, improving the model’s ability to capture complex shapes and scales of Chinese character strokes. The introduction of the Shape-IoU loss function further refined bounding box regression, particularly demonstrating higher precision in locating and segmenting fine strokes. Ablation studies confirmed the specific contributions of these improvements to enhancing model performance. Lastly, the application of Grad-CAM++ technology in generating segmentation prediction heat maps supported the model’s interpretability and visual analysis, revealing the focal points of the model in the fine segmentation of Chinese character strokes.Experimental validation using the CCSE dataset showed that the YOLOv8n-seg-CAA-BiFPN model outperformed the original YOLOv8n-seg model in terms of average accuracy, average recall rate, and mean average precision in fine segmentation of Chinese character strokes, achieving 84.71%, 83.65%, and 80.11%, respectively. Compared to the YOLOv8n-seg model, these represent improvements of 3.50%, 4.42%, and 3.11%, respectively.

In summary, the YOLOv8n-seg-CAA-BiFPN model not only demonstrated exceptional detection performance in the task of fine segmentation of Chinese character strokes but also maintained high efficiency in model size and recognition speed, meeting the practical application demands. This provides new research directions and practical value for the application of deep learning in Chinese character design and computer-aided design technologies.

## Figures and Tables

**Figure 1 sensors-24-03480-f001:**
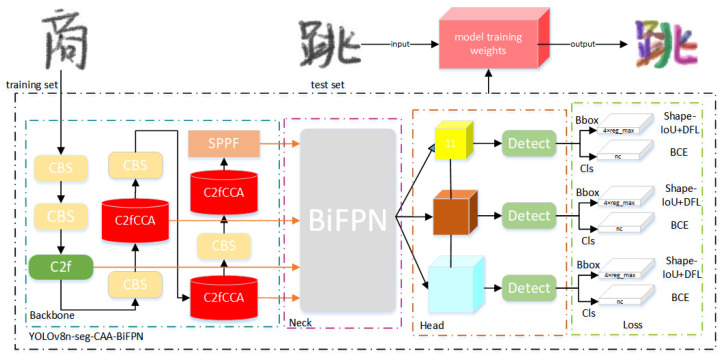
YOLOv8n-seg-CAA-BiFPN Structure.

**Figure 2 sensors-24-03480-f002:**
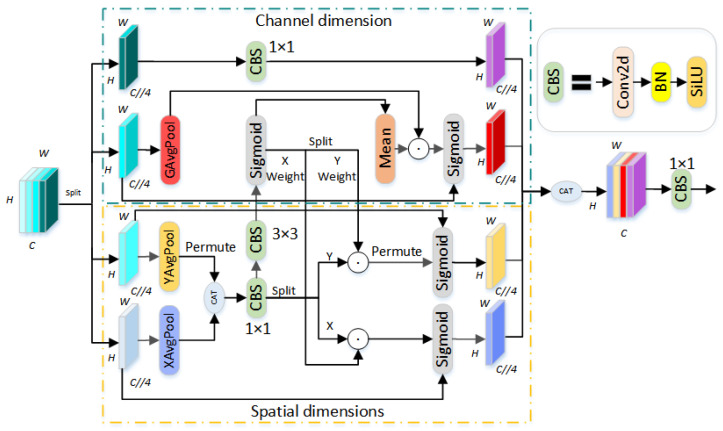
Coordinate Aware Attention Mechanism Structure.

**Figure 3 sensors-24-03480-f003:**
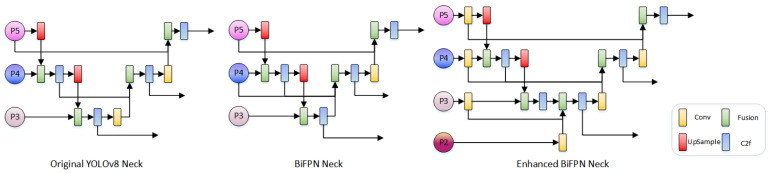
Neck module structure.

**Figure 4 sensors-24-03480-f004:**
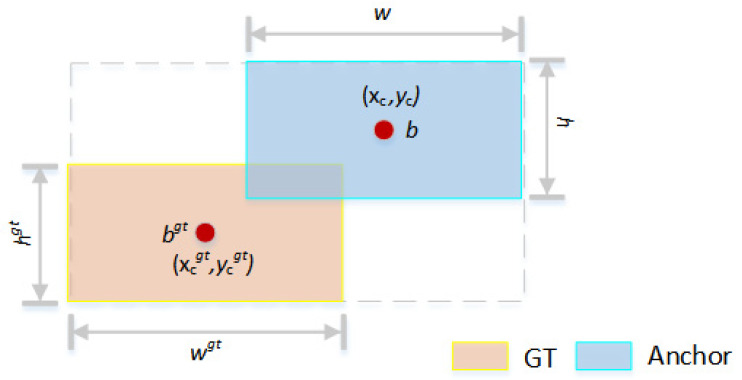
Shape-IoU Schematic.

**Figure 5 sensors-24-03480-f005:**
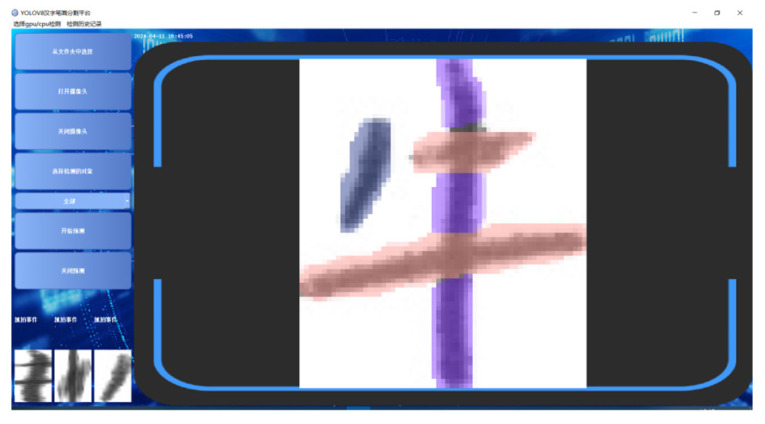
Chinese Character Stroke Segmentation System User Interface.

**Figure 6 sensors-24-03480-f006:**
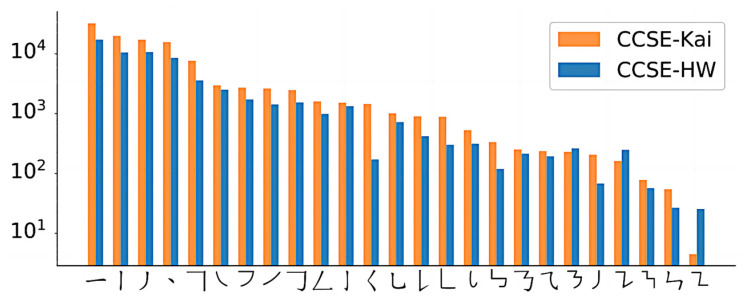
Number of Annotated Instances in CCSE-HW and CCSE-Kai.

**Figure 7 sensors-24-03480-f007:**
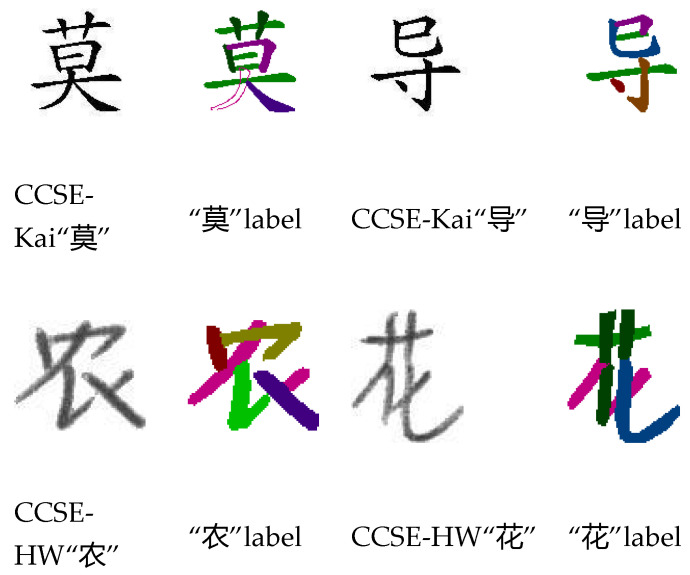
The situation of the CCSE dataset.

**Figure 8 sensors-24-03480-f008:**
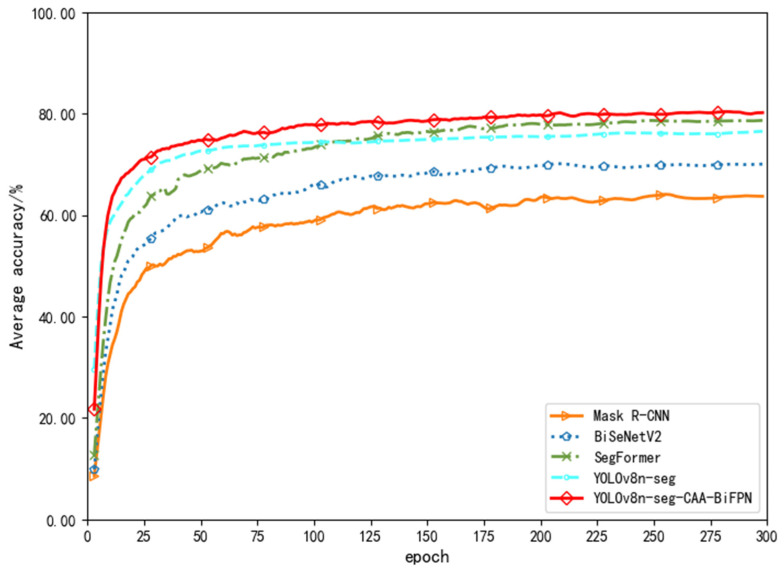
Comparison Chart of Average Accuracy.

**Figure 9 sensors-24-03480-f009:**
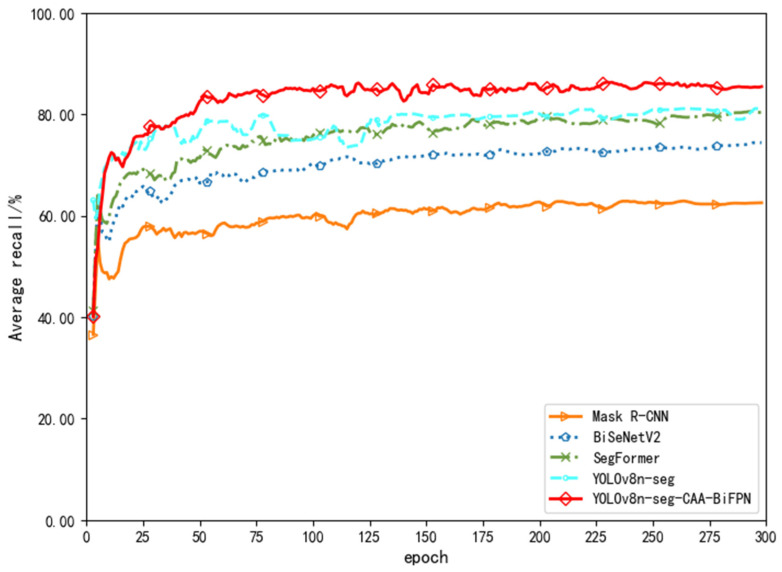
Comparison Chart of Average Recall Rate.

**Figure 10 sensors-24-03480-f010:**
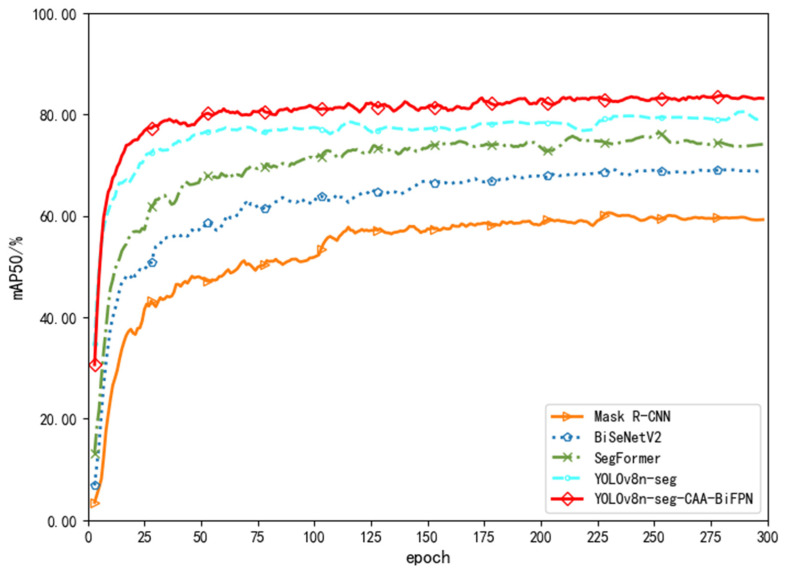
Comparison Chart of Mean Average Precision.

**Figure 11 sensors-24-03480-f011:**
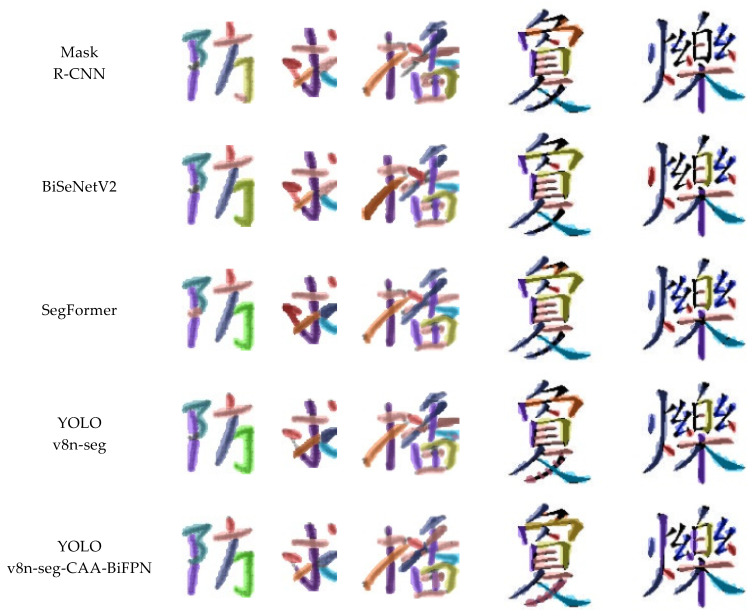
Comparison of segmentation results between different models.

**Figure 12 sensors-24-03480-f012:**
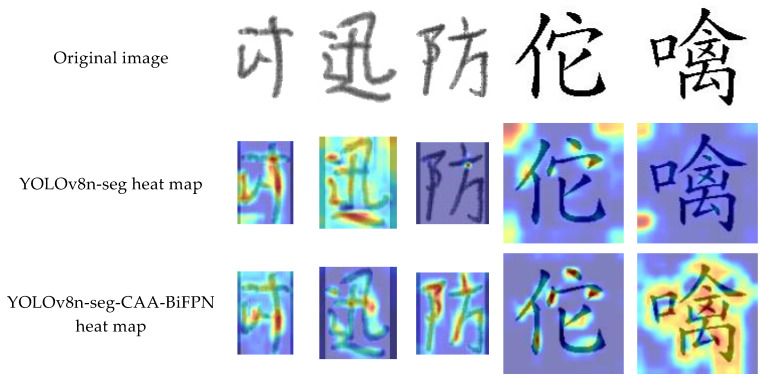
Comparison of heat map between different models.

**Table 1 sensors-24-03480-t001:** Comparative Metrics of Different Models.

Model	Average Accuracy/%	Average Recall/%	mAP/%	Weight/MB
Mask R-CNN	62.66	58.73	60.78	233.0
BiSeNetV2	74.15	68.01	72.09	20.3
SegFormer	80.36	74.33	76.09	15.4
YOLOv8n-seg	81.21	79.23	77.00	6.5
YOLOv8n-seg-CAA-BiFPN	**84.71**	**83.65**	**80.11**	**4.8**

**Table 2 sensors-24-03480-t002:** Results of ablation experiment.

CAA	Enhanced BiFPN	Shape-IoU	Average Accuracy/%	Average Recall/%	mAP/%	Detection Time t/s
			81.21	79.23	77.00	0.029
√			83.12	81.01	78.90	0.030
	√		82.78	80.64	78.54	**0.025**
		√	82.02	79.83	77.62	0.029
√	√	√	**84.71**	**83.65**	**80.11**	0.027

**Table 3 sensors-24-03480-t003:** Ablation Study of Attention Mechanisms in YOLOv8n-seg.

Model	Average Accuracy/%	Model Size/%	Detection Time t/s
YOLOv8n	77.00	6.5	0.029
YOLOv8n+SE	77.12	6.8	0.029
YOLOv8n+CA	77.25	6.8	0.029
YOLOv8n+CBAM	77.51	6.9	0.030
YOLOv8n+CAA1	77.91	6.7	0.029
YOLOv8n+CAA2	77.84	6.6	0.029
YOLOv8n+CAA3	77.74	6.6	0.029
YOLOv8n+CAA	78.90	6.8	0.030

**Table 4 sensors-24-03480-t004:** Ablation Study Results for Enhanced BiFPN in YOLOv8n-seg.

Model	Average Accuracy/%	Model Size/%	Detection Time t/s
YOLOv8n	77.00	6.5	0.029
YOLOv8n+BiFPN	77.83	4.2	0.024
YOLOv8n+Enhanced BiFPN	78.54	4.5	0.025

## Data Availability

The datasets used or analyzed during the current study will be available from the corresponding author upon reasonable request.

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
