# Peer review of "Fine Segmentation of Chinese Character Strokes Based on Coordinate Awareness and Enhanced BiFPN"

_sensors, 2024, doi:10.3390/s24113480_

Round 1

Reviewer 1 Report

Comments and Suggestions for Authors

This study proposes an improved scheme based on the YOLOv8n-seg model, named YOLOv8n-seg-CAA-BiFPN, to enhance the accuracy and adaptability of traditional models in segmenting Chinese character strokes. Through theoretical demonstration and experimental analysis, this improvement is evident in defining edges and processing diverse fonts. It also mitigates overfitting issues caused by sample imbalance. However, some minor issues should be described more accurately or clearly.

1. The first letter in line 166 should be lowercase.

2. Please carefully check the description of the correspondence between Fig. 2 and the demonstration. It seems somewhat vague, and the correlation between the text and the figure is not clear enough.

3. The explanation of the proposed Enhanced BiFPN is also too vague, especially the enhanced weighted fusion strategy in line 207.

Author Response

Respected Experts,

Firstly, I would like to express my deepest gratitude for the invaluable comments and suggestions you provided during the review process. Your professional guidance is crucial for improving my paper, allowing me to more clearly recognize the shortcomings in my research and how to enhance it to meet higher standards.

In response to your comments: “The first letter in line 166 should be lowercase,” “Please carefully check the description of the correspondence between Fig. 2 and the demonstration. It seems somewhat vague, and the correlation between the text and the figure is not clear enough,” and “The explanation of the proposed Enhanced BiFPN is also too vague, especially the enhanced weighted fusion strategy in line 207.” I have made the necessary corrections to lowercase the first letter in line 166 and refined the description of formula 9 at line 155 to “formation through global average pooling, Gav,” aligning it better with the scholarly expression format. I have detailed the description and corrected the grammar related to Fig. 2, clarifying the CAA structure's workflow and algorithmic implications. For the Enhanced BiFPN section, I have added formula derivation and further explanations to help readers better understand the working principles of the structure.

Thank you once again for your valuable time and professional advice. I look forward to your review of my revisions and further guidance and suggestions. I am confident that with your assistance, my research will significantly improve and contribute to the field.

Yours sincerely.

Reviewer’s Comments: (1) Regarding the comment “The first letter in line 166 should be lowercase,” Following your reminder, I have corrected the lowercase issue in line 166 and refined the description of formula 9 at line 155 to “formation through global average pooling, Gav,” to better align with the scholarly expression format.

(2) Regarding the comment “Please carefully check the description of the correspondence between Fig. 2 and the demonstration. It seems somewhat vague, and the correlation between the text and the figure is not clear enough,” Following your reminder, I have made detailed introductions and grammatical corrections to the text describing Fig. 2, adding subject-verb and clause sequencing to better express the specific workflow and the meaning of the CAA structure’s algorithm.

(3) Regarding the comment “The explanation of the proposed Enhanced BiFPN is also too vague, especially the enhanced weighted fusion strategy in line 207,” Following your reminder, I have added formula derivation and detailed explanations to the Enhanced BiFPN fusion section to better help readers understand the working principles of the structure.

Reviewer 2 Report

Comments and Suggestions for Authors

Faced with significant complexity of Chinese characters, the authors rely on the YOLOv8n-seg model for Chinese character segmentation and recognition, bringing many improvements. In this way, the model uses the Cooperative Aware Attention (CAA) mechanism which allows the model to adjust its focus adaptively by accurately calculating and integrating attention weights in different spatial dimensions. When dealing with very complex scene segmentation tasks, the PAN structure may lead to insufficient feature fusion in the YOLOv8n-seg model. To solve this problem, the authors propose an improved BiFPN structure as a replacement for the neck architecture. Comprehensive analysis shows that the improved YOLOv8n-seg model significantly improved the performance and stability of Chinese character stroke fine segmentation tasks.

The study is well conducted, clear and understandable

However, we can criticize the lack of comparison with existing systems dealing with the same problem in order to correctly situate the displayed performances of the final model.

Author Response

Respected Experts,

Firstly, I would like to express my deepest gratitude for the valuable feedback and suggestions provided during the review process. Your professional guidance is immensely valuable in improving my article, helping me recognize the shortcomings of my research and guiding me on how to enhance it to meet higher research standards.

In response to your comment, “However, we can criticize the lack of comparison with existing systems dealing with the same problem in order to correctly situate the displayed performances of the final model,” I have made the comparisons of segmentation models more explicit in the manuscript. I have chosen to compare the Mask R-CNN, as selected in the CCSE paper, with other mainstream segmentation models like BiSeNetV2, SegFormer, YOLOv8n-seg, and YOLOv8n-seg-CAA-BiFPN. Being the first to design experiments with the CCSE segmentation dataset and the first to cite it, I opted to thoroughly compare the use of various popular attention mechanisms that have been improved in other similar research papers, along with conducting ablation studies.

Thank you once again for your valuable time and expert advice. I look forward to your review of my revisions and further guidance and suggestions. I am confident that with your help, my research can be significantly enhanced and contribute meaningfully to the field.

Yours sincerely.

Reviewer's Comments: (1) Regarding the comment, “However, we can criticize the lack of comparison with existing systems dealing with the same problem in order to correctly situate the displayed performances of the final model,” following your reminder, I have made the comparisons of segmentation models more explicit in the manuscript. I have chosen to compare the Mask R-CNN, as selected in the CCSE paper, with other mainstream segmentation models like BiSeNetV2, SegFormer, YOLOv8n-seg, and YOLOv8n-seg-CAA-BiFPN. Being the first to design experiments with the CCSE segmentation dataset and the first to cite it, I opted to thoroughly compare the use of various popular attention mechanisms that have been improved in other similar research papers, along with conducting ablation studies.